# The structural coverage of the human proteome before and after AlphaFold

**Eduard Porta-Pardo**[1,2]*, **Victoria Ruiz-Serra**[1,2], **Samuel Valentini**[3], **Alfonso Valencia**[1,4]*

1 Barcelona Supercomputing Center (BSC), Barcelona, Spain, 2 Josep Carreras Leukaemia Research Institute (IJC), Badalona, Spain, 3 Department of Cellular, Computational and Integrative Biology (CIBIO), University of Trento, Trento, Italy, 4 Institució Catalana de Recerca Avançada (ICREA), Barcelona, Spain

* eporta@carrerasresearch.org (EP-P); alfonso.valencia@bsc.es (AV)

**Data Availability Statement:** PDB data was downloaded from https://rcsb.org AlphaFold data was downloaded from https://alphafold.ebi.ac.uk/ Clinvar data was downloaded from https://ftp.ncbi.nlm.nih.gov/pub/clinvar/vcf_GRCh38/clinvar.vcf.gz

## Abstract

The protein structure field is experiencing a revolution. From the increased throughput of techniques to determine experimental structures, to developments such as cryo-EM that allow us to find the structures of large protein complexes or, more recently, the development of artificial intelligence tools, such as AlphaFold, that can predict with high accuracy the folding of proteins for which the availability of homology templates is limited. Here we quantify the effect of the recently released AlphaFold database of protein structural models in our knowledge on human proteins. Our results indicate that our current baseline for structural coverage of 48%, considering experimentally-derived or template-based homology models, elevates up to 76% when including AlphaFold predictions. At the same time the fraction of dark proteome is reduced from 26% to just 10% when AlphaFold models are considered. Furthermore, although the coverage of disease-associated genes and mutations was near complete before AlphaFold release (69% of Clinvar pathogenic mutations and 88% of oncogenic mutations), AlphaFold models still provide an additional coverage of 3% to 13% of these critically important sets of biomedical genes and mutations. Finally, we show how the contribution of AlphaFold models to the structural coverage of non-human organisms, including important pathogenic bacteria, is significantly larger than that of the human proteome. Overall, our results show that the sequence-structure gap of human proteins has almost disappeared, an outstanding success of direct consequences for the knowledge on the human genome and the derived medical applications.

## Author summary

Protein structures are key to understand many biological phenomena at the molecular scale: from the effects of genetic variation to how different proteins interact with each other to create molecular pathways that, together, have a biological function. Obtaining experimental structures, however, is extremely consuming in terms of both, time and resources.

For this and other reasons, scientists have long worked to develop computational approaches that predict the structure of a protein using only its sequence as input.

TCGA data was downloaded from https://api.gdc.cancer.gov/data/1c8cfe5f-e52d-41ba-94da-f15ea1337efc.

**Funding:** E.P-P and V.R-S are supported by the La Caixa Junior Leader Fellowship LCF/BQ/PI18/11630003 from La Caixa Foundation (https://fundacionlacaixa.org/ca/home). E.P-P is also supported by a Ramon y Cajal fellowship from the Spanish Ministry of Science (RYC2019-026415-I). A.V. is supported by Institució Catalana de Recerca Avançada (ICREA - https://www.icrea.cat) The funders had no role in study design, data collection and analysis, decision to publish, or preparation of the manuscript.

**Competing interests:** The authors have declared that no competing interests exist.

Recently, a group of scientists at Deepmind have developed AlphaFold2, a computational tool that is extremely accurate at this task. Moreover, they have used this tool to predict the structures of all human proteins.

In this manuscript we provide an overview of the structural coverage of the human proteome before AlphaFold models were released and how much we have gained thanks to these models. We also show how the gain affects our understanding of human pathogenic variants, both germline and somatic. Finally, we provide evidence suggesting that the gain in non-human organisms is larger than for the human proteome, particularly in the case of bacteria.

## Introduction

Ever since the first protein structure was published in 1958 [1] it was clear that structure information is essential to understand the biological functions of proteins. In that sense, the last 63 years have witnessed an outstanding progress thanks to several milestones. First and foremost, the Protein Data Bank [2] has emerged, since its creation in 1971, as the key biological database to organize and standardize experimental protein structural data. Also, several groups in the late 1980s and early 1990s observed that the protein structure was much more conserved than its sequence [3], which led to the creation of the first computational tools to predict protein folding [4–8]. These tools are extremely important, as the growth of protein sequence data has far outpaced that of experimentally-determined protein structures. In order to systematically assess the performance of all these tools and monitor the advances of the protein folding prediction field, the Critical Assessment of protein Structure Prediction (CASP) experiments were established in 1994 [9]. These experiments have held a high-standard in the field and in recent years have witnessed the massive progress that protein structure prediction has made thanks to, among others, the use of artificial intelligence approaches [10].

Protein structures have a wide-array of applications in biotechnology and biomedicine. Understanding the structure of a protein can help in identifying which mutations in enzymes can make them acquire new properties that make them interesting from a biotechnological or environmental perspective [11]. Also, in the case of biomedicine, they can be used to perform *in silico* experiments to compute the affinity of tens of thousands of small molecules or antibodies for proteins associated with different diseases [12]. Further biomedical application of protein structures involves the study of the consequences of genetic variants, both acquired [13] or inherited [14]. For instance, computational tools based on protein structures are often more accurate [15,16] than those based on linear features [17] when prioritizing relevant protein-coding variations in patients [18], a much needed strategy when experimental characterization of millions of mutations is not realistic.

Over the last two editions of CASP, the computational tool AlphaFold [19], developed by DeepMind, has achieved a performance that suggests that the structure prediction of individual protein domains has virtually been solved [20]. Moreover, in conjunction with the EMBL-EBI, the team behind AlphaFold has recently released a database of predicted protein structures for the whole human proteome [21]. These two outstanding accomplishments, together, have generated a great excitement in the scientific community in general and amongst structural biologists in particular, as it opens a wide-array of research possibilities.

Given the profound transformations that structural biology has recently experienced, the importance of protein structures in biomedicine, and the potential transformation that the field might experience in light of the release of AlphaFold code and models for human

proteins, it is important to quantify the contribution of these models in general and of high-quality models in particular. The latter are particularly relevant as they might be able to provide insights into the biochemistry of proteins that had not yet been amenable to experimental structure determination or the effects of mutations in such proteins.

Here we quantified the consequences of the addition of the AlphaFold models to the accumulated knowledge on protein structures and models. Our results show that AlphaFold predictions have significantly increased our coverage for the entire human proteome, particularly in proteins integral to membrane, enzymes involved in lipid metabolism and olfactory receptors. The direct contribution of AlphaFold's protein models to proteins with biomedical interest is more limited, as these proteins were already better characterized.

Overall, AlphaFold increases the structural coverage from 48% to 76% of all human protein residues and, importantly, the coverage with high-quality models (sequence identity with a PDB chain is $\geq$ 50%) from 31% to 50%. Moreover, the number of proteins with no structural coverage at all has dramatically decreased from 5,027 proteins to just 29. Together, the combined data from experimental structures, template-based homology modelling with high sequence identity and high-accuracy models from AlphaFold will open a new era in structural biology in particular and human biomedicine in general.

## Results

### The structural coverage of the human proteome before AlphaFold

Over the last 25 years, the protein structure field has made significant advances. This is evidenced by the fact that the Protein Data Bank (PDB) [2], the main database for protein structure coordinates, in 1995 had 4,455 protein coordinate files, whereas by 2020 the number had increased to 177,806 (Fig 1A). The relative contribution of human proteins (sequence identity of human proteins with PDB chains is >95%) has also increased over these 25 years, now making 27% of all PDB structures, compared to only 17% in 1995 (Fig 1B). These advances, importantly, have translated into a much higher coverage of the human proteome. We quantified it by aligning all the 19,250 protein isoforms of the human protein coding genes from ENSEMBL [22] that also matched exactly UniProt [23] sequences against all the PDB chains (n = 596,542) using BLAST [24]. We considered only hits with an e-value below 1e-8 and sequence identity $\geq$ 20%, the limit thresholds for template-based homology modelling [3].

Additionally, for a more straightforward interpretation of the results, we set in this study 3 different levels of protein structural coverage according to 3 ranges of sequence identity percentages between a protein and a PDB chain. The ranges are: >95%, which includes the real structure of the protein; 95–50%, which contains useful templates to study the impact of mutations [25]; and 50–20%, which contemplates structural coverage of distant related proteins assuming that structure is more conserved than sequence. With this definition our results show that the current structural coverage of the human proteome is 48% of all human protein residues (Fig 1C) when sequence identity $\geq$ 20%. This is significant progress compared to 1995, when the coverage was 7 times less (6.3%). If we are more stringent and consider only the regions of the proteome with a sequence identity >95% with a PDB protein chain, then the current coverage decreases to 17%, but even so this is 33 times higher than in 1995 (0.47%).

Not all the human proteome has a structure (such as in the case of intrinsically disordered proteins [26]) or its structure has been more difficult to experimentally determine (as in the case of membrane proteins). To further explore this issue, we predicted the presence of two types of linear regions: PFAM domains [27], which are evolutionary conserved functional regions, and intrinsically disordered regions (IDRs) which are proteins that do not have unique equilibrium residue coordinates and instead exist either as conformational ensembles

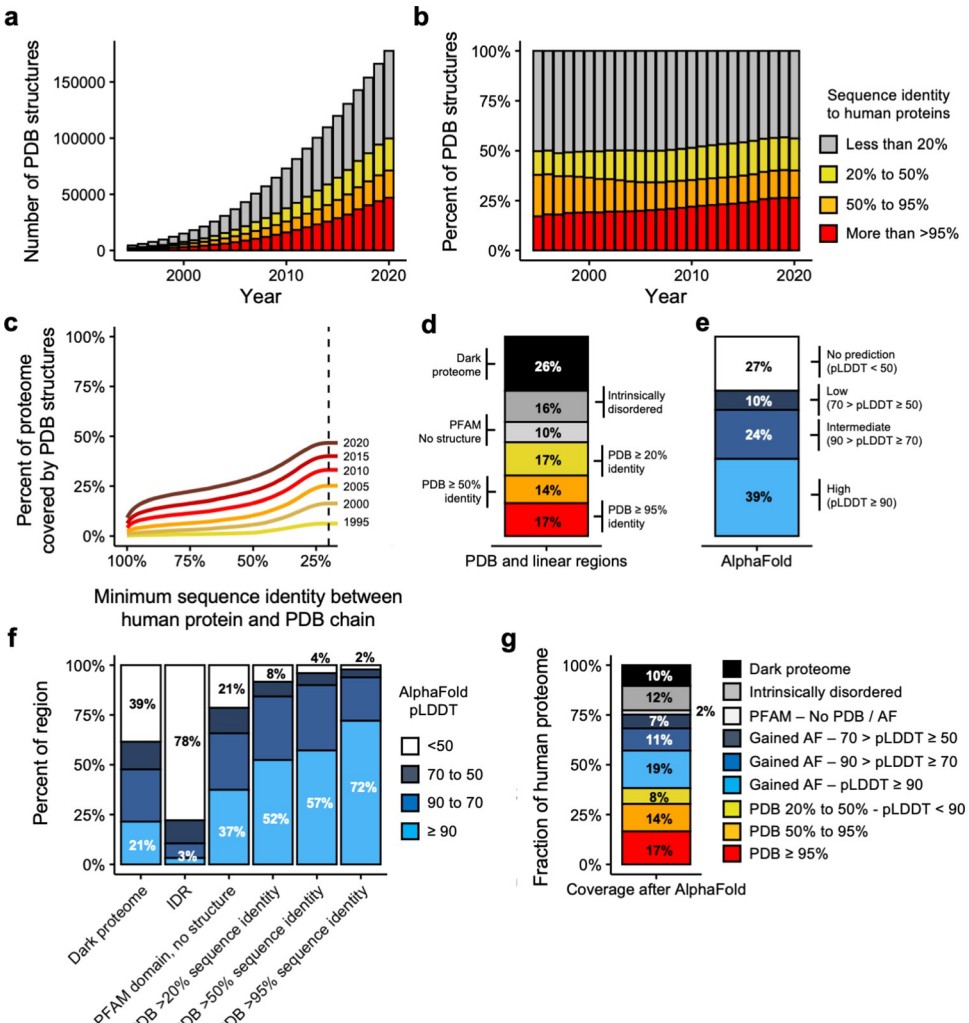

**Fig 1. Current coverage of the human proteome. a-b)** Barplot showing the absolute (**a**) or relative (**b**) number of PDB coordinate files mapping to human proteomes at >95%, 95–50% and 50–20% thresholds of sequence identity. Legends in barplots **a** and **b** are the same. **c)** Evolution of the coverage of the human proteome by three-dimensional coordinate files in the Protein Data Bank (y-axis) according to the minimum percent identity of the BLAST hits (x-axis). Each line represents the coverage using only the coordinate files available in PDB in a given year. **d)** Barplot showing the coverage of the human proteome by different types of structural features, both linear (PFAM domains and IDRs) and three-dimensional (PDB) (y-axis is the same as in **c**). **e)** Coverage of the proteome by different AlphaFold pLDDT score thresholds (y-axis is the same as in **c**). **f)** Coverage (y-axis) of different types of regions (x-axis) depending on AlphaFold confidence levels. **g)** Current coverage (y-axis) of the human proteome.

of multiple macro-states or without any equilibrium 3D structure and with relatively flat probability distribution on allowable 3D structures (**Fig 1D**). The remaining proteins, neither structured nor predicted to be disordered (the dark proteome [28]) were also considered in this analysis (**Fig 1D**). Moreover, it's important to notice that some PDB structures could be IDR in physiological conditions and the structure inferred is the result of the crystallization procedure potentially introducing some misclassifications. Results show that for 10% of all residues of the human proteome there is a PFAM domain assigned but no template available to model a structure, 16% of all residues are predicted to be disordered and we have no features for the remaining 26% of all residues.

In summary, in terms of three-dimensional coverage, 17% of all human protein residues currently have experimental structures (sequence identity with a PDB chain is >95%), 14% can be modelled with good accuracy (sequence identity between 95% and 50%) and 17% can be modelled with reasonable accuracy (sequence identity between 50% and 20%). As for the rest of the proteome, without considering IDRs, 36% of residues are missing structural coverage even though 10% of them belong to a known PFAM domain (**Fig 1D**).

## The structural coverage of the human proteome after AlphaFold

We next analyzed the structural predictions of AlphaFold for the human proteome [21]. AlphaFold has, along with the actual three-dimensional coordinates, a measure of how confident one can be about the prediction itself which is the predicted local distance difference test (pLDDT). According to its developers, a pLDDT above 90 is of high confidence, between 90 and 70 of intermediate confidence, between 70 and 50 of low confidence, and the coordinates of any residue with a pLDDT below 50 should be disregarded [21]. At the residue level, Alpha-Fold models have high confidence predictions for 39% of the human proteome, intermediate-confidence predictions for 24%, low-confidence predictions for 10% and no prediction (i.e. pLDDT < 50) for 27% (**Fig 1E**).

We calculated the overlap between the distribution of linear and template-based homology regions with the AlphaFold prediction confidence scores (**Fig 1F**). As expected, there was a correspondence between the sequence identity between proteins and PDB chains and Alpha-Fold's pLDDT score. While 72% of all residues with sequence identity to a PDB template >95% had a pLDDT above 90, the percent decreased to 52% of all residues with sequence identity between 20% and 50% (**Fig 1F**). Consistently, we observe that 78% of all residues that were predicted to be in IDRs (i.e. disordered) had a pLDDT below 50. Overall these results show consistency between the foldability of a protein region and AlphaFold's pLDDT score.

AlphaFold makes a significant contribution in the portion of the human proteome that lacks structure coverage, i.e. a number of PFAM domains without known experimental structure or template to build homology models and the dark proteome. High-confidence structures (pLDDT >90) are predicted for 37% and 21% of the residues corresponding to PFAM domains without structure or the dark proteome respectively (**Fig 1F**). Still, 21% and 39% of the residues of PFAM domains without structure and the dark proteome are associated with a pLDDT < 50 (**Fig 1F**), which could be indicative of truly disordered regions.

The integration of our previous structural knowledge through experimental structures, template-based homology modeling, PFAM domains, IDRs together with AlphaFold predictions, shows that the current overall coverage of the human proteome at the residue levels is excellent (**Fig 1G**). For the rest of the paper, we consider a region as having a highly accurate protein structure model or prediction if it either has a sequence identity above 50% with a PDB chain or it has an AlphaFold pLDDT score above 90.

With this definition, excellent structures or models are available for 50% of the residues, 31% coming from PDB and 19% from AlphaFold. An additional 26% of the human proteome can be covered with other structural models, either through template based modeling of PDB chains with a sequence identity with the template between 20% and 50% (8%) or through AlphaFold models with intermediate (11%) or low confidence (7%). Approximately 12% of the proteome is predicted as disordered and only 2% of all residues have a PFAM domain without structure nor model. Finally, the fraction of the proteome without any annotation has decreased to only 10%, albeit given the high correlation between regions with pLDDT below 50 and intrinsic disorder, these are likely to also be IDRs, which would put the fraction of disordered proteome to 22%.

## Changes in coverage at the protein level

In this section, we analyse how AlphaFold complements our knowledge at the protein level. When comparing the number of human proteins without structural coverage before and after considering AlphaFold predictions, we observe that the number drops from 5,027 to just 29 proteins (Fig 2A). This means that we currently have either experimental or predicted structural data for some region of 99.8% of all the human proteins we considered (19,250 proteins, **Methods**).

We studied in more detail the proteins for which we previously had no structural data, i.e. the aforementioned 5,027 proteins. A GO term enrichment analysis [29] on these proteins reveals a strong enrichment in olfactory receptors (259 genes, OR > 2.3, p < 7e-26), genes integral to membrane (1,917 genes, OR 1.3, p < 1e-35) or genes without any previous GO annotation (2,897 genes, OR > 1.9, p < 1e-100) among many others.

Notably, AlphaFold models provide structural coverage for over 50% of the protein's length for 4,459 of these 5,027 proteins (**Fig 2B**). While this number decreases to 1,408 proteins if we

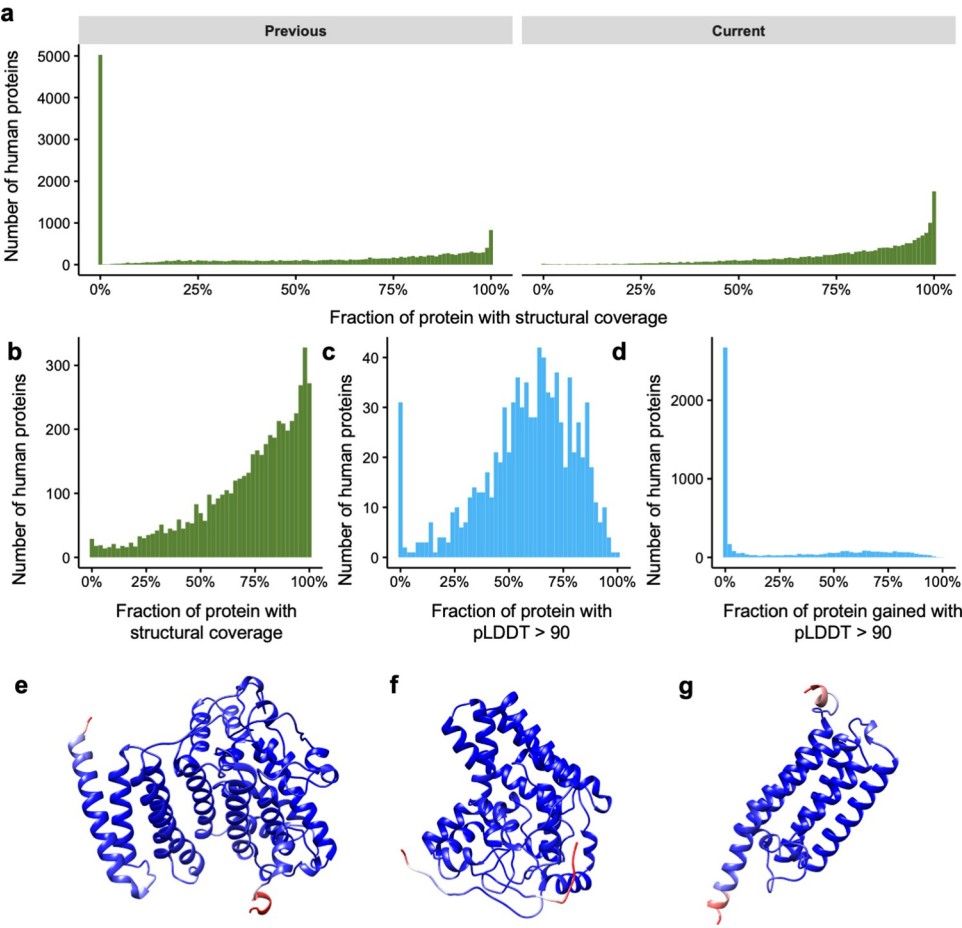

**Fig 2. Changes in the structural coverage at the protein level after AlphaFold. a)** Histogram showing the number of proteins (y-axis) according to their structural coverage (x-axis) before (left) and after (right) the release of AlphaFold models. **b)** Histogram showing the number of proteins for which we previously had less than 1% of structural coverage (y-axis) according to their current structural coverage after AlphaFold. **c)** Same as **b** but now including only high-confidence (pLDDT > 90) AlphaFold predictions (x-axis). **d)** Histogram showing how much AlphaFold high-confidence predictions contribute (x-axis) to our coverage of proteins with >95% structural coverage. **e-g)** AlphaFold models for previously structureless AGMO, DEGS1 and PEMT proteins. Models are colored in blue-red scale showing the pLDDT score for the residue, with red representing low pLDDT and blue high pLDDT.

only consider high-quality predictions (an AlphaFold model with pLDDT > 90 is covering more than half the protein's length, **Fig 2C**), this is still a significant improvement considering that before AlphaFold we had no structural data for these proteins.

Lastly, we quantified how much AlphaFold contributes to the proteins for which our current structural coverage (sequence identity between protein and PDB chain is > 20% or pLDDT > 50) is above 95% of the protein length (n = 5,178, **Fig 2D**). This revealed that, while in most cases we already had a significant amount of structural data, either experimental (sequence identity > 95%) or through template-based homology (sequence identity > 20%), there are 1,448 proteins with very high structural coverage (PDB sequence identity >20% or pLDDT > 90 for more than 95% of all residues of the protein) where AlphaFold contributes over 50% of the structural data. This is the case of relevant proteins in the biomedical context such as AGMO, an enzyme that has been linked to numerous diseases such as cancer and diabetes [30], DEGS1, another enzyme with a critical role in lipid metabolism [31], or PEMT, another lipid metabolism enzyme and was amongst the first genes associated with non-alcoholic fatty liver disease [32] (**Fig 2E–2G** respectively). In all three cases no structural information was known nor any template with enough sequence identity to build a model (sequence identity > 20%). However, AlphaFold predicted with high confidence the structures of most of all these proteins (90%, 95% and 91%, respectively).

## Structural coverage of regions with biomedical interest

Next, we focused on different subsets of genes of biomedical interest, as their protein structures can be useful in the design of new treatments [33] or to understand the consequences of genetic variants [34,35]. For this purpose, we focused on the protein structural coverage of disease related genes and mutations extracted from DisGeNet [36], OncoKB [37], TCGA [15] and ClinVar [38] databases (**Methods**).

Based on experimental PDB structures, the structural coverage of disease genes from DisGeNet (25%), cancer driver genes from TCGA (30%) and OncoKB (28%) or genes related to neurological (34%) or autoimmune (36%) diseases is always higher than the average coverage for the human proteome (17%, **Fig 3A**). This is likely due to an increased effort in determining the structure of disease-associated proteins than for proteins of the rest of the human genome. Therefore, it is not surprising that for these genes, the contribution of AlphaFold is not as high as for the rest of the proteome (**Fig 3A**). Particularly, DisGeNet genes gain between 14% to 28% of structural coverage, depending if we consider only high confidence results (pLDDT >90) or all AlphaFold predictions (pLDDT >50), reaching a total coverage up to 78%. In the case of cancer driver genes, the contribution is lower, as TCGA and OncoKB genes gain 6% and 8% of high confidence structural coverage. Overall, the contribution from AlphaFold with some degree of confidence together with PDB previous knowledge, sets the current high-quality structural coverage for cancer driver genes (TCGA and OncoKB) to 52% and the overall coverage to 70%.

In the context of mutations, we analyzed the structural coverage of Clinvar (**Fig 3B**) considering their annotated pathogenicity. In the case of mutations annotated as "benign", before AlphaFold we had high-quality structural data for approximately 17% to 19%, and AlphaFold models add an additional 12%. For pathogenic mutations, instead, we already had a better experimental structural coverage, between 48% and 58%. That being said, AlphaFold models add an additional 13% of high-quality structural data. Considering all structural annotations, we now have structural data for 94% of all pathogenic variants. Notably, the review status of Clinvar mutations does not affect the conclusions of our analysis (**S1 Fig**). In fact, there seems

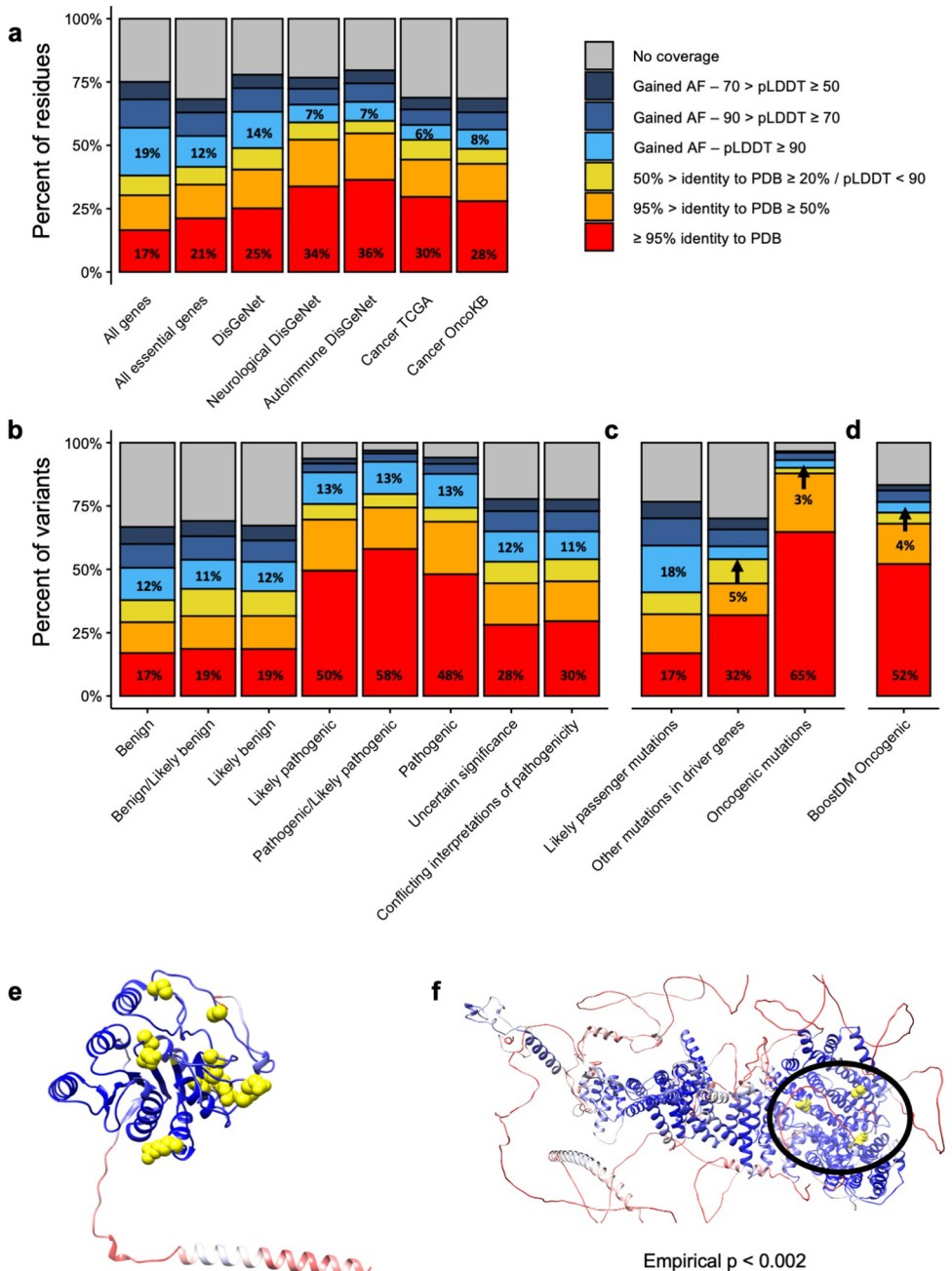

**Fig 3. Changes in structural coverage of biomedical proteins due to AlphaFold models. a)** Current structural coverage (y-axis) of different subsets of proteins (x-axis). Bars are colored according to the source of the structural coverage. **b)** Same as **a** but focusing on Clinvar mutations classified by their pathogenicity (x-axis). **c)** Same as **a** but focusing on somatic mutations from TCGA, classified by their likely oncogenicity (x-axis). **d)** Same as **a** but focusing on oncogenic mutations from BoostDM. **e)** AlphaFold model for B3GALT6. Residues are colored according to their pLDDT from red (lower values) to blue (higher values). Pathogenic mutations from Clinvar are highlighted in yellow. **e)** AlphaFold model for MED12. Coloring is the same as for **d**, but yellow residues indicate oncogenic mutations.

to be a positive correlation between the PDB coverage of pathogenic variants and their review status: all pathogenic variants with four stars have an experimental PDB structure, whereas only 50% of the pathogenic variants with zero stars do.

In the case of TCGA (**Fig 3C**), mutations were classified according to whether they were predicted to be either oncogenic, located in cancer driver genes or located in any other protein (likely passengers). Results resemble those observed in the Clinvar dataset. The experimental structural coverage of likely passenger mutations is exactly the same as for the entire proteome, 17%, and high-quality AlphaFold models add an additional 12%. In the case of driver genes, however, we started from a much higher experimental structural coverage (**Fig 3C**), as we had experimental structures for 32% of the non-oncogenic mutations in these genes, and Alpha-Fold high-quality models only add an additional 5%. For mutations predicted to be oncogenic in these genes [15], we already had experimental structures for 65% of them, and AlphaFold only added 3%. We observed the same trend for another recent set of predicted oncogenic mutations by *in silico* mutagenesis of cancer driver genes [39], with experimental structural data from PDB covering 52% of all oncogenic mutations and AlphaFold only adding 4% (**Fig 3D**). It should be noted, though, that the algorithms predicting the oncogenicity of both sets of somatic mutations use in part structural information, so the results are likely biased towards regions with pre-existing structural data.

In summary, although the additional contribution of AlphaFold models to pathogenic and oncogenic mutations is apparently low, it still provides important structural context for some mutations. This is the case of B3GALT6, a gene without structural information and that, when mutated, can cause a Mendelian disease. Thanks to the high-confidence structural model provided by AlphaFold, it is now possible to appreciate a 3D clustering of the disease-causing mutations in this gene (**Fig 3E**). Similarly, oncogenic mutations in the cancer driver gene MED12 previously had no structural data at all, but are now covered by a high-confidence model. In fact, the mapping of the three oncogenic somatic mutations in MED12 to the Alpha-Fold model reveals that, despite being over 600 aminoacids apart (positions 521, 879 and 1138), they form a cluster together according to mutation3D [40] (p < 0.002, 10,000 bootstrapping iterations), suggesting that they potentially share an oncogenic mechanism (**Fig 3F**).

## Comparison with other species

As we have shown before (**Fig 1B**), protein structures are particularly biased towards human proteins. For this reason we hypothesized that the impact of AlphaFold models could be higher in other organisms. To explore this hypothesis we calculated the structural coverage before and after AlphaFold protein models were released for four organisms: *A. thaliana*, *S. cerevisiae*, *S. aureus* and *M. tuberculosis*. We selected these four organisms because they cover both, prokaryotes and eukaryotes and also because they include well-studied model organisms (*A. thaliana* and *S. cerevisiae*) as well as important bacterial pathogens from a biomedical perspective (*S. aureus* and *M. tuberculosis*).

To evaluate the gain in structural coverage we calculated the sequence identity between all the proteins in these organisms with all the chains in experimental structures from PDB and established the same thresholds of identity as in **Fig 1**. Then, we calculated how many residues were covered by AlphaFold models at different pLDDT thresholds and quantified the gain of structural coverage as in **Fig 1.** Our results show that the structural coverage of these other four organisms provided by either experimental structures of high-quality template-based homology modeling was lower than for the human proteome (**Fig 4A**). The most dramatic example is *A. thaliana*, which only had experimental structures for less than 2% of all the residues in its proteome. Next, we estimated how much high- or intermediate-quality (pLDDT > 90 and pLDDT > 70 respectively) structural coverage is provided by AlphaFold models in these organisms. In all four cases, AlphaFold provides more structural coverage than for the human proteome (**Fig 4B**). The detailed results per organism can be found in **S2**

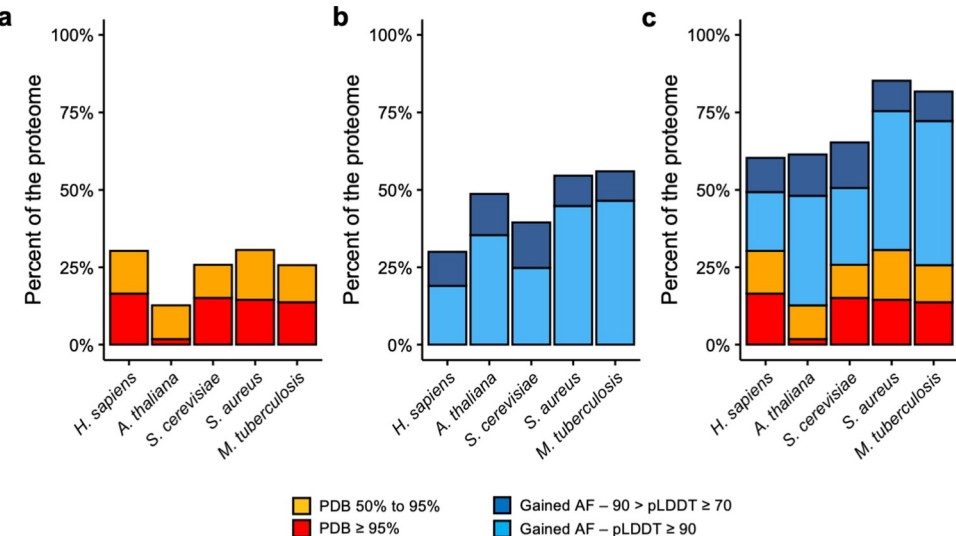

**Fig 4. Changes in protein structural coverage in other organisms. a)** Comparison of the structural coverage (y-axis) of the five different organisms (x-axis) based on PDB sequence identity. **b)** Additional structural coverage provided by AlphaFold models in the different species, split by pLDDT score. **c)** Current high quality structural coverage of the five organisms combining PDB and AlphaFold data.

**Fig**. Notably, when combining both sources of structural data, the final coverage of the human proteome with high- or intermediate quality structural data is lower (60%) than for the other four organisms, which reach coverages of 81% and 85% for *M. tuberculosis* and *S. aureus* respectively (**Fig 4C**).

In summary, the improvement of the structural coverage using AlphaFold models in non-human organisms is likely to be even higher than for the human proteome. Here we have presented four good examples that benefit from AlphaFold models data and that, in the case of the prokaryotic organisms (*S. aureus* and *M. tuberculosis*), they almost cover 100% of the residues, or, in the case of eukaryotic organisms (*S. cerevisiae* and *A. thaliana*), they reach similar structural coverage results to those observed in human.

## Discussion

In this study we quantified the contribution of AlphaFold protein models to our already considerable knowledge of the human proteome. Before AlphaFold models were released we had high-quality structural data for a significant fraction of the human proteome (31%), either through experimental structures (17% of all residues) or through homology models built with templates of high sequence identity (14% of all residues). However, there was still a large fraction of the proteome for which, despite enough evidence about its foldable potential (i.e. proteins with good representation in terms of the corresponding PFAM), we had no structural data.

The models from AlphaFold have increased our high-quality coverage by an additional 20%, putting it at half of all human protein residues. Moreover, they have virtually closed the gap of structureless foldable regions, as we now have some structural coverage (either through AlphaFold or PDB low sequence identity models) for 75% of all human residues, including almost all PFAM domains. Regarding the latter, another important dataset that has been recently published are the structural models predicted with RoseTTAFold, another artificial intelligence tool, for all PFAM domains [41]. In fact, our preliminary analysis shows that

RoseTTAFold tends to predict the same exact structure as AlphaFold models with high pLDDT [25]. Finally, approximately 23% of all human residues are likely to be disordered, which puts the fraction of the human proteome that can potentially have a structure but for which we have no models or templates at only 2%, a remarkable achievement.

In terms of individual proteins, while before AlphaFold we had no structural coverage at all for 5,027 proteins, this number has been reduced to only 1,499 if one considers only proteins with high-quality AlphaFold models (pLDDT > 90 in more than 10% of the protein) and to 29 if one includes all AlphaFold predictions. Importantly, these previously structureless proteins for which we now have high-accuracy models include several important genes in human disease such as DEGS1, ARID2 or B3GALT6.

Our results also suggest that most of the genuinely new contributions of AlphaFold to the structural coverage are in genes that are olfactory receptors, proteins integral to the membrane or, most importantly, genes that have no Gene Ontology annotations. However, one must also bear in mind that some of these predictions will not be correct, either due to actual errors of the software or, in other cases, due to missannotations of public databases such as in the case of CRIPAK [42]. On the positive side, we also expect AlphaFold predictions to enlighten the function of many of the proteins for which we previously had no GO annotations. Since we now have high-confidence predictions for, at least, some parts of many of these proteins, it should be easier to identify protein domains and begin to assign functions and biological processes to these proteins.

In the specific case of disease-associated genes, our high-quality coverage before AlphaFold models were released was higher (between 25% and 36% of all residues) than the average of the human proteome (17%). This could be, among other reasons, because disease-associated genes are more studied than the rest, as evidenced by their increase in their relative representation in PDB: in 1995 only 2% of all PDB structures were from human cancer genes, whereas by 2020 the fraction has increased to 4%. Therefore, the contribution of AlphaFold to this group of proteins is more limited, between 5% and 14% if one considers high quality models, compared to 19% for all residues in the human proteome.

In the case of mutations the bias is even higher, not only because disease-associated mutations are more studied, but also because they tend to be located in protein regions that form structures. For this reason AlphaFold models only provide high-confidence structural details for an additional 13% and 3% for Clinvar pathogenic and TCGA oncogenic mutations respectively. That being said, AlphaFold models can be used to generate hypotheses behind the pathogenic mechanisms of some of these mutations where we previously had no structure data at all, as we have exemplified for MED12.

The impact of AlphaFold models is clearly not restricted to the human proteome, but also extensible to other organisms. While the detailed study of each of them is out of the scope of the present manuscript, we have provided evidence showing that the contribution of AlphaFold models to coverage of non-human organisms is even higher than for the human proteome, at least in the case of *M. tuberculosis*, *A. thaliana* and *S. cerevisiae*. This is likely to be even more important for bacteria because, unlike eukaryotes, most of their proteome is folded and the experimental structural data is oftentimes lacking. As we have shown for *S. aureus* and *M. tuberculosis*, the structural coverage after including AlphaFold models reaches almost 100%, with most of the structural data coming from AlphaFold models of high quality. Based on these results, we believe that an extremely important contribution of AlphaFold models to human biomedicine will be in the fight against antibiotic-resistant bacteria, for example by facilitating new virtual screening analyses that will now be able to cover almost 100% of the proteome of these bacteria.

In conclusion, we now have either experimental data, template-based homology models or artificial intelligence-derived models for virtually all foldable protein regions in the human proteome (76% of the proteome). Moreover, 50% of the human proteome is covered with experimental or high-quality structural data (more than 50% sequence identity to PDB or more than 90 pLDDT for AlphaFold models). While at the end of 2020 we were already in a very good position (31% of the proteome), AlphaFold has greatly contributed in closing the gap of protein structure coverage, providing details in protein families that were not amenable to other structure determination approaches. Overall, we have practically completed our coverage of the human foldable proteome, an outstanding achievement that will open the door to more challenging problems in structural biology in general and in biomedicine in particular, such as the prediction of protein-protein interaction complexes [43–45], the relative positioning of protein domains in multi-domain proteins [46], the identification of immunogenic peptides [47] or the prediction of the consequences of different types of mutations.

## Methods

### Homology search

A BLAST [24] search was performed against 596,542 PDB chains extracted from a total of 253.444 protein bioassemblies deposited in PDB by the date of download (04/03/2021) using the human proteome as a query (Ensembl v103) [24]. We only used one protein isoform per gene, the one that matched exactly the protein sequence from the reference Uniprot human proteome. In the end we used 19,250 protein sequences. Significant hits between ENSEMBL protein sequences and PDB chains were those with e-value lower than 1e-8 and sequence identity percent greater or equal to 20%.

### PFAM domains

We used Pfamscan [48] to identify PFAM domains [27] in the 19,250 ENSEMBL / Uniprot sequences. The database of PFAM-A models was downloaded on June 29th 2021 and created on March 19th 2021. We only kept those PFAM domains identified with an e-value below 1e-8.

### IUPRed2

To identify intrinsically disordered regions we used IUPRed2 [49] (downloaded on April 17th 2021). We used the "long" disorder option and considered as disordered all those residues with a score above 0.5. We used IUPRed2 because, based on recent benchmarkings, it was a good compromise between speed and accuracy [50].

### AlphaFold human models

AlphaFold models for human proteins were downloaded on July 23rd 2021 from https://alphafold.ebi.ac.uk. We extracted the sequences and compared them to the ENSEMBL protein sequences that matched the UniProt reference human proteome used for the PDB analysis. For comparison purposes all the analyses and results presented here are based on the subset of 19,250 protein sequences for which the ENSEMBL, Uniprot and AlphaFold protein sequences were identical. We also extracted pLDDT values for each residue from the AlphaFold models, as these are stored as if they were the B-factor of the protein coordinates file [19,21].

## Analysis of non-human models

AlphaFold models for non-human organisms were downloaded on November 15th 2021 from https://alphafold.ebi.ac.uk. We followed the same exact procedure as for the analysis of human proteins.

## Gene lists

The list of "All essential genes" includes all those with a pLI score [51] above 0.9. Disease-associated gene annotations come from DisGeNet [36]. The list of "DisGeNet" genes from Fig 3A consists of all DisGeNet genes with a score above 0.45 and where "diseaseType" is "disease". The list of "Neurological DisGeNet" genes includes all genes from DisGeNet where the score is above 0.45 and the diseaseName is in the following list: "Alzheimer_s Disease", "Amyotrophic Lateral Sclerosis", "Bipolar Disorder", "Schizophrenia", "Depressive disorder", "Mental Depression", "Autistic Disorder", "Major Depressive Disorder" and "Parkinson Disease". The list of "Autoimmune DisGeNet" genes includes all genes from DisGeNet where the score is above 0.45 and the diseaseName is in the following list: "Asthma", "Diabetes Mellitus, Non-Insulin-Dependent", "Lupus Erythematosus, Systemic", "Rheumatoid Arthritis", "Crohn Disease", "diabetes Mellitus, Insulin-Dependent", "Ulcerative Colitis" and "Psoriasis". The list of "Cancer TCGA" genes includes all those found as drivers in the PanCancerAtlas analysis [15]. "Cancer OncoKB" includes all genes annotated as cancer genes in OncoKB [37] (downloaded on July 26th 2021).

## Variant datasets

Variant files from TCGA and ClinVar datasets were retrieved from public repositories (https://api.gdc.cancer.gov/data/1c8cfe5f-e52d-41ba-94da-f15ea1337efc and https://ftp.ncbi.nlm.nih.gov/pub/clinvar/vcf_GRCh38/clinvar.vcf.gz). The protein location of all variants was predicted using the Ensembl Variant Effect Predictor (VEP [52]; version 98.3). Annotations regarding the clinical significance of Clinvar variants were extracted from the Clinvar file. Oncogenic mutations from TCGA include all those predicted to be oncogenic in [15]. The list of driver genes used in Fig 3D is from OncoKB. Mutations from BoostDM [39] were obtained from the IntoGen website (www.intogen.org).

## Software

All statistical analyses were done using R 4.0.2. Graphical plots were create with the packages "ggplot2" [53], "pacthwork" and "reshape2". Molecular graphics and analyses performed with UCSF Chimera [54], developed by the Resource for Biocomputing, Visualization, and Informatics at the University of California, San Francisco, with support from NIH P41-GM10331.

## Supporting information

**S1 Fig.** Structural coverage of Clinvar mtuations (y-axis) depending on their pathogenicity (different panels) and their review status (x-axis).
(TIFF)

**S2 Fig.** Structural coverage (y-axis) of the proteome of the four different organisms before (left) and after (right) including the AlphaFold models.
(TIFF)

## Acknowledgments

The authors would like to thank the DeepMind team for sharing their models of human proteins.

## Author Contributions

**Conceptualization:** Eduard Porta-Pardo, Alfonso Valencia.

**Data curation:** Eduard Porta-Pardo.

**Formal analysis:** Eduard Porta-Pardo, Victoria Ruiz-Serra, Samuel Valentini.

**Funding acquisition:** Eduard Porta-Pardo.

**Investigation:** Eduard Porta-Pardo, Victoria Ruiz-Serra.

**Methodology:** Eduard Porta-Pardo.

**Project administration:** Eduard Porta-Pardo.

**Resources:** Eduard Porta-Pardo.

**Supervision:** Eduard Porta-Pardo, Alfonso Valencia.

**Validation:** Eduard Porta-Pardo.

**Visualization:** Eduard Porta-Pardo.

**Writing – original draft:** Eduard Porta-Pardo, Victoria Ruiz-Serra, Alfonso Valencia.

**Writing – review & editing:** Eduard Porta-Pardo, Victoria Ruiz-Serra, Samuel Valentini, Alfonso Valencia.

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
