## [Decision Letter · Decision Letter 0]

20 Oct 2021

Dear Mr Porta-Pardo,

Thank you very much for submitting your manuscript "The structural coverage of the human proteome before and after AlphaFold" for consideration at PLOS Computational Biology.

As with all papers reviewed by the journal, your manuscript was reviewed by members of the editorial board and by several independent reviewers. In light of the reviews (below this email), we would like to invite the resubmission of a significantly-revised version that takes into account the reviewers' comments.

The comments by the reviewers were generally positive, with a few (mainly technical) issues remaining to be fixed. It has been argued that the paper would benefit from using more organisms for a comparative assessment of the distribution of classes. One reviewer also notes that the code used to generate the data is not available.

It has been noted that, while the paper was under review, the authors have contributed to a preprint which covers roughly the same topic. See: https://www.biorxiv.org/content/10.1101/2021.09.26.461876v1

In light of this, the authors **should argue why they believe this submission to be different from their contribution to that paper**. Expanding the scope of the paper, e.g. by adding more organisms or using additional data, would be an obvious answer. The final call on this will have to be made by the Editorial Office according to journal policy.

We cannot make any decision about publication until we have seen the revised manuscript and your response to the reviewers' comments. Your revised manuscript is also likely to be sent to reviewers for further evaluation.

Sincerely,

Silvio C. E. Tosatto, PhD

Guest Editor

PLOS Computational Biology

Arne Elofsson

Deputy Editor

PLOS Computational Biology

The comments by the reviewers were generally positive, with a few (mainly technical) issues remaining to be fixed. It has been argued that the paper would benefit from using more organisms for a comparative assessment of the distribution of classes. One reviewer also notes that the code used to generate the data is not available.

It has been noted that, while the paper was under review, the authors have contributed to a preprint which covers roughly the same topic. See: https://www.biorxiv.org/content/10.1101/2021.09.26.461876v1

In light of this, the authors should argue why they believe this submission to be different from their contribution to that paper. Expanding the scope of the paper, e.g. by adding more organisms, would be an obvious answer. The final call on this will have to be made by the Editorial Office according to journal policy.

Reviewer's Responses to Questions

**Comments to the Authors:**

Reviewer #1: Porta-Pardo et al. in the article “The structural coverage of the human proteome before and after AlphaFold” provided an analysis aiming at describing the state-of-the-art about our current capacity of inferring the structure of the human proteome. They compared and measured experimentally-derived structures, template-based homology models and artificial intelligence predictions as provided recently by the AlphaFold2 method and available in AlphaFoldDB.

The manuscript is of interest as it can provide a good reference for all discussions about the available knowledge of human protein structures. The language is simple and clear and the methodology used is fair. However, there are a couple of issues that need to be clarified or modified and that are critical for the presented statistics and in my opinion prevent acceptance of the manuscript in the current form.

Major

My main concern is about the definition of disordered residues which is provided by the IUPred-long tool. That choice is never justified nor is it discussed that the tool tends to underpredict disorder. The authors mentioned CASP results to justify the use of AlphaFold over other prediction tools, however they didn’t do the same for disorder predictors. They could cite the recent Critical Assessment of Intrinsic protein Disorder (CAID) and mention that IUPred is a good compromise between speed and accuracy even though it isn’t in the top ten (PMID: 33875885).

In terms of coverage of the human proteome, a statistic is already available from the following MobiDB page, https://mobidb.org/statistics?proteome=UP000005640. The page provides a comparison with other disorder prediction tools and takes the same proteome reference used by AlphaFoldDB (Proteome ID UP000005640) which includes all expressed human proteins.

The second point is about the definition of the human proteome dataset. The choice of using ENSEMBL proteins and selecting one isoform for each gene is fine but somehow it complicates the reasoning and reproducibility. UniProt offers a great service in managing and cross-referencing ENSEMBL and other databases. Under the human proteome page (https://www.uniprot.org/proteomes/UP000005640) it is possible to retrieve all human proteins which are expressed (78,120 proteins, UniProt version 2021_03). Also, at the same page UniProt provides a link to download representative proteins, one protein per gene (20,600 proteins). The number of representative proteins is very close to the one used by the authors. I suggest using the UniProt definition of the proteome (20,600 representative proteins) in order to be consistent with AlphaFoldDB (https://alphafold.ebi.ac.uk/download). I’m pretty sure that it would not change the overall results.

Minor

In the abstract it is not clear whether AlphaFold promotes to structure a fraction of disordered residues or the actual “dark residues”, or both. In other words it is not clear why the structured residues (75%) does not sum up with the 22% of the dark proteome.

Figure 2. I suggest explaining why the 3 example structures have been selected. That is well explained in the manuscript but not in the caption. Also showing what are the residues which were not covered by any structure before AlphaFold would help to understand why the method was useful for those cases. In the end the pLDDT is not that useful for those examples as they are almost completely predicted with a very high score.

Reviewer #2: The paper by Eduard Porta-Pardo etal is a conceptually straightforward but still well executed exploration of how AlphaFold has changed our ability to have good structural models in the human proteome. It adds to the work done by the AlphaFold team and importantly provides a third party view of this, with a focus on disease causing genes as well as overall statistics.

There is nothing major which causes an issue or concern here (ie, the science is solid), however, I think this sort of exploration would be far more powerful is extended to other organisms with AlphaFold predictions, as I think the contrast between the human scenario (with more experimental focus) and other organisms would be important. As I suspect the big issue will be on just importance/impact of this paper, I think this will make for a more rounded and impactful paper.

Minor issues:

I think the authors could have introduced their >95%, 50-95% and 20-50% identity buckets better in the text; One has to draw lines arbitarily somewhere, but they should be up front about this in the text as well as in the figures. This is more stylistic than anything else.

Another point, the authors use the European convention of . for separator with thousands, whereas I think plos expects the US/UK convention of comma.

Reviewer #3: AlphaFold is a deep learning algorithm developed by Google's DeepMind that predicts protein structure from its amino acid sequence. The AlphaFold database containing the predictions for the human proteome and 20 other organisms has now been released to the community. This paper evaluates the impact of the AlphaFold database on our knowledge of human proteins. It also evaluates the impact on the knowledge of disease-associated genes and mutations. Lastly, the authors provide examples of some disease-associated proteins where AlphaFold provided key insights. They conclude that the AlphaFold has largely closed the gap between the sequence-structure knowledge gap.

Strengths:

i) Extensive evaluation of the coverage of human proteome before and after AlphaFold. Databases used - PDB, ENSEMBL

* The amount of protein for which no structure information was available has reduced from 53% before AllphaFold (Fig. 1d) to 25% after AlphaFold (Fig. 1e)

* AlphaFold reduced the number of proteins without structural coverage from 4,832 to only 29

* GO enrichment analysis on these 4,832 proteins revealed strong enrichment in 269 olfactory receptor genes, 66 lipid metabolism genes, 1700 genes integral to membrane, and 1056 genes without previous information.

ii) Extensive evaluation of the improvement of our knowledge of the disease related genes and mutations extracted from - DisGeNet, OncoKB, TCGA, ClinVar

* The coverage of disease-releated genes based on their PDB structures (50-61%) is higher than that of the average proteome (47%). Thereforefore, it is not surprising that AlphaFold does not contribute as much to them as it does to the rest of the proteome

* AlphaFold increased the percentage of high quality coverage proteins from 31% to 43% for benign mutations of ClinVar, and from 71% to 83% for pathogenic mutations

ii) Good examples of the disease-causing genes B3GALT6, MED12, in which AlphaFold has revealed the specific-clustering due to disease-causing mutations

The authors conclude that the after AlphaFold has "practically completed" our coverage of the human foldable proteome, and this shall open the door for more challenging problems in structural biology.

Weakness:

i) Parts of the paper of were very hard to follow and need relatively extensive re-writing and tightening

ii) The mutations taken from ClinVar were reported as benign or pathogenic. The authors need to provide criteria they used to select those variants. Normally, variants with at least one star would be selected. Otherwise, the results could be polluted by noise.

**Have the authors made all data and (if applicable) computational code underlying the findings in their manuscript fully available?**

Reviewer #1: **No: **The code is not available.

Reviewer #2: Yes

Reviewer #3: Yes

PLOS authors have the option to publish the peer review history of their article (what does this mean?). If published, this will include your full peer review and any attached files.

Reviewer #1: No

Reviewer #2: No

Reviewer #3: No
---

## [Decision Letter · Decision Letter 1]

7 Jan 2022

Dear Mr Porta-Pardo,

We are pleased to inform you that your manuscript 'The structural coverage of the human proteome before and after AlphaFold' has been provisionally accepted for publication in PLOS Computational Biology.

Best regards,

Silvio C. E. Tosatto, PhD

Guest Editor

PLOS Computational Biology

Arne Elofsson

Deputy Editor

PLOS Computational Biology

Reviewer's Responses to Questions

**Comments to the Authors:**

Reviewer #1: The authors responded to all my answers and modified the manuscript. I think it is now suitable for publication

Reviewer #2: The authors have responded to my concerns well, and I appreciate the new section on other species (it would be churlish to ask for more, but I wonder if they should do figure 4 over everything in a years time!).

It is a sound paper, and the other changes have improved it.

Reviewer #3: The authors have provided good responses. I noticed this time an additional issue that I hope the authors can fix before this paper proceeds to production.

*) IDR are defined as "protein regions predicted not to have any tertiary structure in their native state". I think this definition is somewhat loose because disordered regions do have 3D structure, it's just that it's time-changing. IDRs may also exist as conformational ensembles such that multiple macro-states are simultaneously present in a population of molecules and one cannot obtain a unique fixed 3D structure using traditional X-ray crystallography or NMR spectroscopy. The authors can hopefully improve on their definition, perhaps something like "IDRs do not have unique equilibrium residue coordinates and instead exist either as conformational ensembles of multiple macro-states or without any equilibrium 3D structure and with relatively flat probability distribution on allowable 3D structures".

* Wrt IDR, a number of protein structures in PDB might be disordered when under physiological conditions. Multiple proteins have been crystalized under conditions of high salt, with binding partners, etc. but in fact may be disordered when in isolation. The paper cannot fix these problems, but this might be something to mention as limitations because this work does not focus on monomers only.

**Have the authors made all data and (if applicable) computational code underlying the findings in their manuscript fully available?**

Reviewer #1: Yes

Reviewer #2: Yes

Reviewer #3: Yes

PLOS authors have the option to publish the peer review history of their article (what does this mean?). If published, this will include your full peer review and any attached files.

Reviewer #1: No

Reviewer #2: No

Reviewer #3: No

---

## [Editor Report · Acceptance letter]

20 Jan 2022

PCOMPBIOL-D-21-01536R1 

The structural coverage of the human proteome before and after AlphaFold

Dear Dr Porta-Pardo,

I am pleased to inform you that your manuscript has been formally accepted for publication in PLOS Computational Biology. Your manuscript is now with our production department and you will be notified of the publication date in due course.

With kind regards,

Agnes Pap
